# A Prospective Comparative Evaluation of Handheld Ultrasound Examination (HHUS) or Automated Ultrasound Examination (ABVS) in Women with Dense Breast

**DOI:** 10.3390/diagnostics12092170

**Published:** 2022-09-08

**Authors:** Nicole Brunetti, Sara De Giorgis, Simona Tosto, Alessandro Garlaschi, Giuseppe Rescinito, Barbara Massa, Massimo Calabrese, Alberto Stefano Tagliafico

**Affiliations:** 1Radiology Section, Department of Health Sciences (DISSAL), University of Genova, Via L.B. Alberti 2, 16132 Genoa, Italy; 2Department of Radiology, IRCCS—Ospedale Policlinico San Martino, Largo Rosanna Benzi 10, 16132 Genoa, Italy; 3Cyto-Histopathological Unit, IRCCS—Ospedale Policlinico San Martino, Largo Rosanna Benzi 10, 16132 Genoa, Italy

**Keywords:** breast cancer, breast density, ultrasound, mammography, automated breast ultrasound

## Abstract

Mammography is the gold standard examination for breast cancer screening. In women with high breast density, mammography has reduced sensitivity. In these women, an additional screening option is often recommended. This study prospectively compared ABVS and HHUS in women with mammography-negative examinations and dense breasts. Materials and methods: N = 222 women were evaluated prospectively and consecutively between January 2019 and June 2019 (average age 53 years; range 39–89). McNemar’s test and ROC analysis were used with standard statistical software. We included in the study both symptomatic and asymptomatic women with dense breasts. Women included underwent both HHUS and ABVS after mammography with independent reading. Results: N = 33/222 (15%) women resulted in having breast cancer. Both ABVS and HHUS identified more cancers than standard mammography, and both HHUS and ABVS had false-positive examinations: n = 13 for HHUS and n = 12 for ABVS. We found that HHUS had better accuracy than ABVS. The AUC of the ROC was 0.788 (95% CI 0.687–0.890) for ABVS and 0.930 (95% CI 0.868–0.993) for HHUS. This difference was statistically significant (*p* < 0.05). Conclusions: HHUS was more accurate in breast cancer detection than ABVS. Multicentric studies must confirm these data for supplemental imaging in women with dense breasts.

## 1. Introduction

Mammography is a fundamental diagnostic and screening tool for detecting breast cancer (BC). Breast density has a masking effect on lesion identifications; in women with high parenchymal density, there is a lower sensitivity to mammography [1,2]. Moreover, these women have an increased intrinsic risk of developing breast cancer. For these reasons, further investigations are often recommended, and today handheld ultrasound examination (HHUS) is a possible option [3,4]. Automated ultrasound examination (ABVS) is a relatively recent implementation of HHUS. HHUS is believed to be an operator-dependent examination requiring a high level of skill and experience. However, ABVS has been implemented to overcome some HHUS disadvantages, such as reproducibility and the possibility to evaluate images acquired by technicians remotely. Indeed, the radiologist reports the ABVS exam after the acquisition and does not perform it. For these reasons, this method could be beneficial, especially in the screening setting. Brunetti et al. reported that the time required by radiologists to report ABVS examination is longer than for HHUS, even considering the exam’s interpretation time [5]. Multiple studies have demonstrated similar sensitivity, cancer detection rate, diagnostic accuracy, and image quality for both ABUS and HHUS [6,7,8]. However, ABVS has some limitations, such as a higher rate of false-positives and recall than HHUS [9].

One of the ABVS peculiarities is that it provides better detection of architectural distortions using the so-called “coronal view”; spiculated lesions with desmoplastic retraction may appear on the coronal plane as ‘a retraction phenomenon sign’, which is highly suspicious for malignancy [10,11].

Therefore, this study aims to compare ABVS and HHUS in a cohort of women with mammography-negative examinations and dense breasts to understand the advantages or disadvantages of the ABVS method in this particular setting. In this study, HHUS and ABVS were performed with independent reading.

## 2. Materials and Methods

### 2.1. Patients

Two hundred twenty-two women, with an average age of 53 years (39–89 years), were recruited consecutively between January 2019 and June 2019 in a comparative prospective trial between tomosynthesis (DBT) and HHUS (NCT03033030) for a clinical diagnostic study to compare HHUS to ABVS. Women included after HHUS underwent ABVS on the same day. In our research, breast radiologists interpreted HHUS or ABVS images independently. This is a subsidiary study that included both symptomatic and asymptomatic women, including women with clinical findings or high-risk patients with BIRADS breast density C and D [12]. We enrolled patients in a mixed screening/symptomatic assessment center following the daily workflow (screening mammograms or women arriving urgently with clinical findings, for example). About 20% of the enrolled patients were symptomatic. BI-RADS breast density was visually assessed as previously reported [13,14] by a radiologist with at least twenty years of experience in breast imaging. Pregnant, lactating, and unable to express valid informed consent were not enrolled in the study. None of the 222 eligible women declined ABVS after HHUS.

### 2.2. HHUS Examination

HHUS was performed by one radiologist with at least fifteen years of experience in breast imaging. HHUS was carried out using a linear transducer at 10 MHz. In all the patients after HHUS, women underwent ABVS. During our HHUS examination, we analyzed every segment of the breast (upper-outer, lower-outer, lower inner, and upper-inner) both in radial and antiradial planes, followed by the study of the nipple and axilla. If necessary, the patient was asked to turn on her side to better study the outer segments of the breast. All the recorded images were sent to PACS.

HHUS examinations were interpreted according to the BI-RADS classification. An assessment of BI-RADS categories U1–3 was considered an adverse finding; instead, BIRADS U4-U5 lesions underwent US-guided core needle biopsy.

### 2.3. ABVS Execution

The ABVS examination was performed with ACUSON S2000 (Siemens, Erlangen, Germany). This tool presents both the 3D (ABVS) and 2D (HHUS) probes, allowing easy execution of both exams. This ABVS system gives a 3D probe with a 15 cm wide FOV transducer. The transducer can automatically scan a deep of 6 cm and an area of 16.8 cm. Image thickness was 0.5 mm. In addition, the 3D probe presents harmonic tissue imaging, a dynamic tissue enhancement, and an automatic brightness correction of the image.

The patients were placed in the supine position with arms above the head, on the cot, by radiology technicians dedicated to breast radiology after a dedicated training of 15 days. First, they placed a hypoallergenic lotion on the breast to aid an acoustic coupling. Then, the high-frequency transducer is positioned on the breast, and compression is carried out. The radiology technicians report the nipple location’s on the screen. The ABVS examination includes three-mayor scans (antero-posterior, lateral and medial) for 1 min. Women should breathe slowly and should not move during the exam. We can choose different cup sizes according to the size of the breast. There is the chance to use a different sequence to cover the entire breast in case of a large cup breast (with cup D, D+). In accordance with cup size, probes are used at different frequencies (A = 11 MHz, B = 10 MHz, C = 9 MHz, D = 9 MHz, D+ = 9 MHz). Additional acquisitions were performed in case of detection of artifacts during scanning. ABVS was successfully performed on all 222 women. The nipple location was used to rebuild images, especially for the coronal view. After capture, images were analyzed on a dedicated workstation in an axial, coronal, and sagittal view using 3D data reconstruction.

### 2.4. ABVS Interpretation

Images were evaluated on a dedicated workstation and reconstructed at a 1-mm thickness. Our interpretation protocol first reads the coronal view and then analyzes axial and sagittal images. Two radiologists with at least ten years of experience in breast imaging read the ABVS images, blinded to HHUS examination. These radiologists underwent a training period in ABVS interpretation of a minimum of four months.

Those lesions with US characteristics of malignancy (BIRADS U4-U5) were recalled.

### 2.5. Histopathology

All patients with BIRADS U4/U5 at US examination underwent US-guided core needle biopsy. At least one year of follow-up with mammography and HHUS was guaranteed for all patients. This follow-up was also guaranteed for patients with benign lesions (BIRADS U1/U3).

### 2.6. Statistical Analysis

We evaluated participants by comparing independently acquired data of HHUS and ABVS, considering excision histology as the reference standard. At least one year of follow-up was guaranteed for all patients. HHUS and ABVS images were interpreted independently by two breast radiologists with at least ten years of experience in breast imaging, blinded to the HHUS findings if they were reporting the ABVS. Prior mammograms or tomosynthesis were available to report HHUS or ABVS as in standard clinical practice. As planned in the trial, according to McNemar’s test, a significance level of 5% was acceptable for the study. ROC analysis compared HHUS and ABVS against histology as the reference standard, and the *p*-value was adjusted using Sidak’s method and considered significant with *p* < 0.05. N = 150 patients were considered statistically and clinically sufficient for a <5% probability of alfa error. Statistical analysis was done with commercially available software (STATA MP 17.0 StataCorp LLC 4905 Lakeway Drive, College Station, TX, USA).

## 3. Results

None of the 222 eligible women declined ABVS after HHUS. N = 33/222 (15%) women had cancer detected by ABVS and/or HHUS and negative mammography. Cancer characteristics are the following: twenty-five invasive ductal carcinoma, three invasive lobular carcinomas, one invasive mucinous carcinoma, and one invasive oncocytic carcinoma, with medium radiological size of 15.7 mm (6–30 mm). In addition, we found three carcinomas in situ (two ductal carcinomas in situ and one solid papillar ductal carcinoma in situ) and five high-risk (B3) lesions: three atypical ductal hyperplasias, one flat epithelial atypia, and one radial scar. B3 lesions we found underwent surgical excision but were not included in the malignant lesions in the data analysis.

Both methods identified more cancers, and both HHUS and ABVS had false-positive examinations n = 13 for HHUS and n = 12 for ABVS.

Instead, on 33 cancers detected, we had n = 2 false-negative examinations for HHUS and n = 13 for ABVS. Significantly, among n = 13 ABVS false-negative exams, two invasive lobular carcinoma and two triple negative ductal invasive carcinomas were only detected by HHUS. On the other hand, ABVS alone detected two hormone-receptor-positive breast cancer.

HHUS examination had higher sensitivity and specificity than ABVS. Indeed, the sensitivity of HHUS was 90.62% (95% CI 75.0–98.0) and 68.75% for ABVS (95% CI 50.0–83.9) for HHUS.

HHUS also presents higher specificity than ABVS: 96.32% (95% CI 92.6–98.5) of HHUS versus 90.62% (95% CI 85.6–94.3) of ABVS.

So we found that the HHUS examination had better accuracy than ABVS. Indeed, the AUC of the ROC was 0.788 (95% CI 0.687–0.890) for ABVS and 0.930 (95% CI 0.868–0.993) for HHUS. The difference in the AUC (Figure 1) between the two methods was statistically significant (*p* < 0.05). Figure 1 shows the ROC curve for HHUS and ABVS.

Other significant tools we calculated:-The positive likelihood ratio for ABVS is 7.33 (95% CI 4.5–12.1)-The positive likelihood ratio for HHUS is 24.6 (95% CI 11.8–51.3)

Table 1 shows the diagnostic performance of ABVS and HHUS examinations.

Some examples of breast cancers detected by HHUS or ABVS are presented in Figure 2, Figure 3, Figure 4, Figure 5 and Figure 6.

## 4. Discussion

In this study, we evaluated the diagnostic performance of ABVS compared to HHUS for breast cancer detection in women with dense breasts and negative mammography. The rationale for using HHUS or ABVS is that mammography has a lower sensitivity in women with dense breasts than those with non-dense breasts. In women with elevated breast density, there is a well-known masking effect on breast cancer [15]. In the comparative trial of Adjunct Screening With Tomosynthesis or Ultrasound in Women With MG-Negative Dense Breasts (ASTOUND-2), adjunct ultrasound had an incremental cancer detection rate of 4.9/1000 screens [3,4]. ABVS is a technique that has been developed to overcome the limitations of operator dependency and lack of reproducibility in HHUS. In addition, ABVS detects distortion thanks to the coronal view [16] and to the retraction phenomenon, which is rarely seen in benign breast lesions [17]. In our study, breast radiologists interpreted HHUS or ABVS images independently. We identified 33/222 cancer using ultrasound (HHUS and/or ABVS) in women with dense breasts and negative mammography. We included in the study both asymptomatic and symptomatic women: also including women with clinical findings or high-risk patients. HHUS had higher accuracy than ABVS; however, both methods identified more cancers than mammography, as expected, but this difference was statistically significant (*p* < 0.05). In several studies, ABVS presents similar diagnostic accuracy and detection rate of HHUS [8,18,19,20,21]. In other studies, ABVS had significantly higher specificity and positive predictive value than HHUS [22,23]. This is the first study in literature where ABVS had a worse diagnostic performance than HHUS. Other studies found that ABVS was at least equal to HHUS [24]. First, this could be caused by the different cohort of patients included in our study (symptomatic and asymptomatic women) that differs from other studies reported in the literature [8]. Clinical data were available to the radiologists performing HHUS or interpreting ABVS. Still, one of the advantages of HHUS is the possibility of performing a clinical examination and patient questioning before the execution of ultrasound.

We acknowledge that medical radiologists with sub-specialization in breast imaging performed HHUS. In contrast, ABVS images were acquired by one technician dedicated to breast imaging and then read by a radiologist, as usually happens in Europe. It is possible that the data of the present paper could be partly fitted with other medical realities (for example, the United States) where technicians or sonologists execute both HHU and ABVS. In our opinion, the results of this study favor the added value of HHUS, especially in a clinical setting. Indeed, using HHUS, the radiologist can also perform patient questioning and clinical examination. HHUS and ABVS could be considered reliable tools with some potential in the screening setting. Both techniques can increase cancer detection, especially in patients with negative mammography and dense breasts. In addition, ABVS is essential when vital barriers to implementing screening with ultrasound are represented by staffing. For this reason, ABVS could be a vital tool for the additional breast cancer screening because the technician performs the execution and the reading by radiologists.

In addition, we acknowledge that our breast radiologists have a long experience with HHUS [3,4], and on the other side, they have a shorter experience with ABVS. Conversely, in several institutions, HHUS is performed by sonologists or technicians, not breast radiologists. Reader experience could favor HHUS examination, so we are not sure that these data could be exportable in other centers with a lower experience in breast ultrasound. ABVS may play a more significant role in areas where technicians conduct ultrasound examinations or where there are no experienced breast radiologists. As Xi Lin et al. [22] reported, we cannot evaluate the performance of ABVS in the screening setting, where the proportion of breast cancer cases is much lower than that in our study. An essential value of our work is represented by the solid study design with independent reading, not always present in others’ ABVS papers [19]. In our study, HHUS and ABVS present a similar rate of false-positive examination (n = 13 for HHUS, n = 12 for ABVS). As reported in the literature [25], also in our experience, the most common reason for the false-positive exam for ABVS (Figure 5) is artefactual shadowing of the nipple; instead, for HHUS, false-positive examinations are due to the detection of a small benign nodule, mainly when associated to a clinical finding. One advantage of ABVS is the better evaluation of the extent of the disease with precise detection of satellite lesions measuring less than 1 cm [26] (Figure 4). In addition, we received n = 2 false-negative examinations for HHUS and n = 13 for ABVS. Significantly, among n = 13 ABVS false-negative exams, two invasive lobular carcinomas and two triple negative ductal invasive carcinomas were only detected by HHUS. Only ABVS detected two hormone-receptor-positive breast cancer thanks to the architectural distortion in the coronal view. In our experience, the most frequent reason for ABVS false-negative was the nipple artifact. Most lesions were in the retro areolar region, obscured by this artifact. Other lesions missed by ABVS were small and peripheral lesions. As reported in the literature, ABVS presented a reduced diagnostic performance for detecting peripherally situated lesions, particularly in the large breasts, with respect to HHUS [27]. Another reason for ABVS’s false-negative examination was the lack of diagnosis of small and aggressive breast cancer. As reported in the literature, in large imaging volumes, small invasive cancers may be easily missed in ABVS, especially small aggressive cancers that are less frequently accompanied by architectural distortion [28], as shown in our case (Figure 6). We reported a summary table with the leading causes of ABVS false-negative examinations (Table 2).

On the other hand, we received two false-negative results with HHUS due to small nodules not detected by the radiologist.

This study has some limitations. First, there is a relatively low number of patients even if the results are statistically significant due to the elevated number of cancer found in this clinical cohort. The number of cancer is consistent with previous studies comparing HHUS and tomosynthesis [4]. The training period for radiologists and technologists was relatively short, reflecting a sub-optimal experience in ABVS compared to the long-lasting experience in HHUS. Finally, the reference standard was biopsy histopathology.

## 5. Conclusions

In conclusion, both HHUS and ABVS identified more cancers in the mammography, as expected. However, HHUS had higher diagnostic accuracy than ABVS. The different results could be because of the fundamental importance of the clinical relationship in symptomatic women that only HHUS can provide. Multicentric studies are required to confirm the comparison data between ABVS and HHUS for supplemental imaging in women with dense breasts.

## Figures and Tables

**Figure 1 diagnostics-12-02170-f001:**
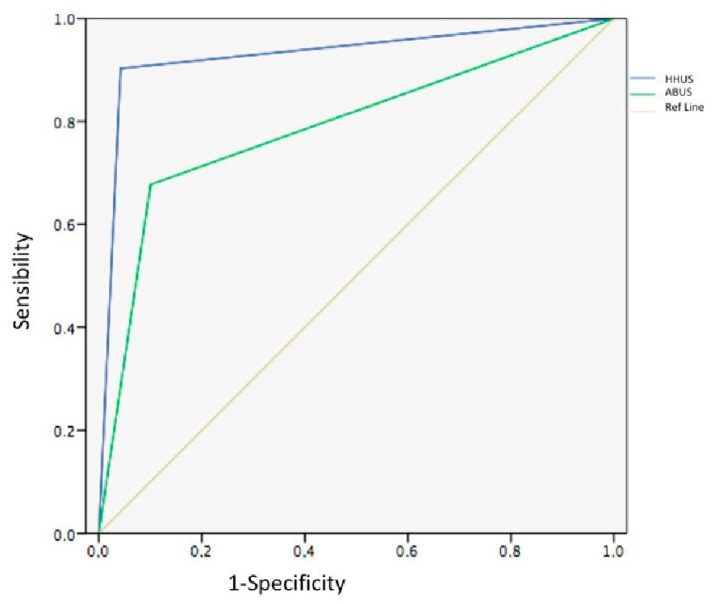
ROC curve. HHUS examination presents a better accuracy than ABVS (*p* < 0.05).

**Figure 2 diagnostics-12-02170-f002:**
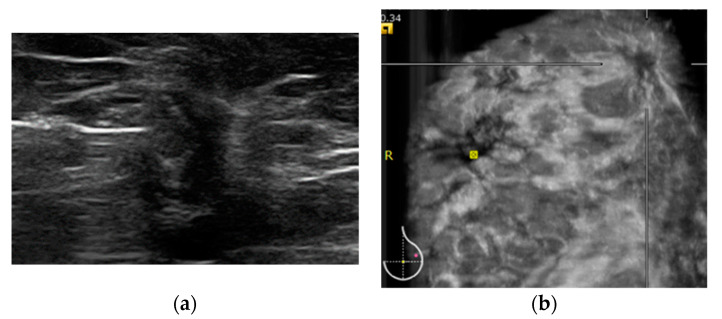
51-year-old woman comes to our institute for routine screening. We reported cancer (invasive lobular carcinoma) identified by both HHUS and ABVS. (**a**) HHUS shows an architectural distortion at the upper-outer quadrant of the left breast. Images (**b**) show the retraction phenomenon in ABVS coronal view.

**Figure 3 diagnostics-12-02170-f003:**
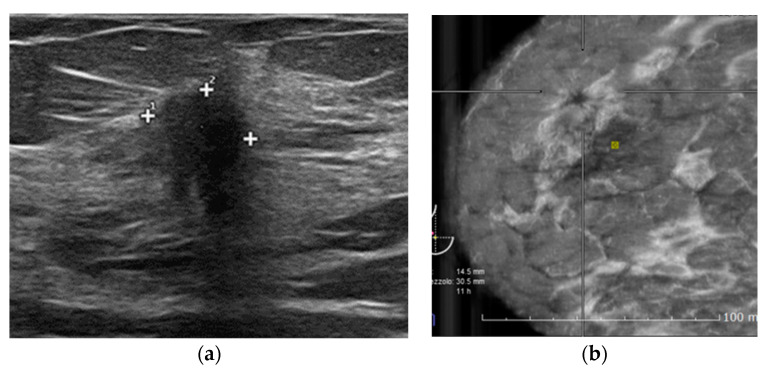
49-year-old woman with no family history of breast cancer performs her annual control. Both HHUS and ABVS identified breast cancer. (**a**) HHUS shows a mass with indistinct margins biopsy-proven invasive ductal carcinoma in the lower outer quadrant of the right breast. (**b**) Retraction phenomenon in ABVS coronal view.

**Figure 4 diagnostics-12-02170-f004:**
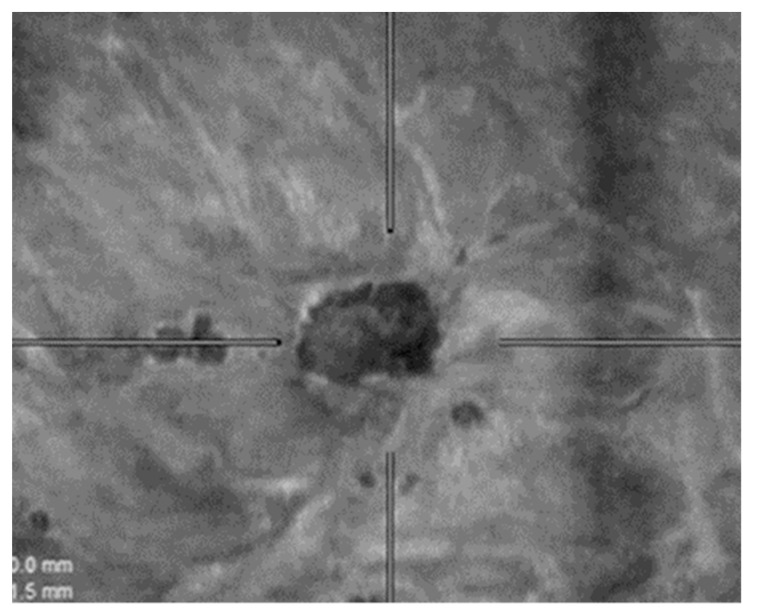
63-year-old woman with a main nodule (invasive ductal carcinoma biopsy-proven). Other smaller lesions (with a size from 5 to 8 mm) are seen in ABVS and not visualized on conventional HHUS.

**Figure 5 diagnostics-12-02170-f005:**
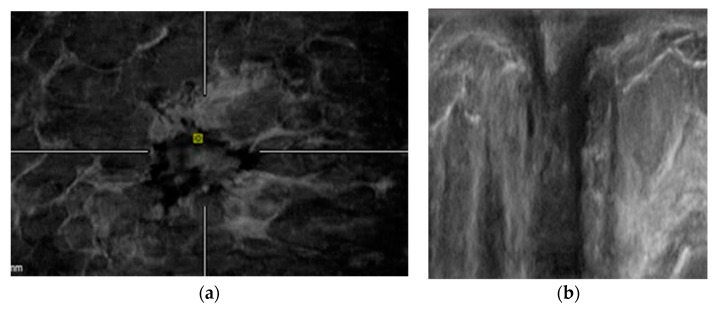
A false-positive case from ABVS caused by nipple artifact. Routine screening for a 67-year-old woman. (**a**) small architectural distortion is seen on ABVS coronal view near the nipple. (**b**) Negative HHUS examination.

**Figure 6 diagnostics-12-02170-f006:**
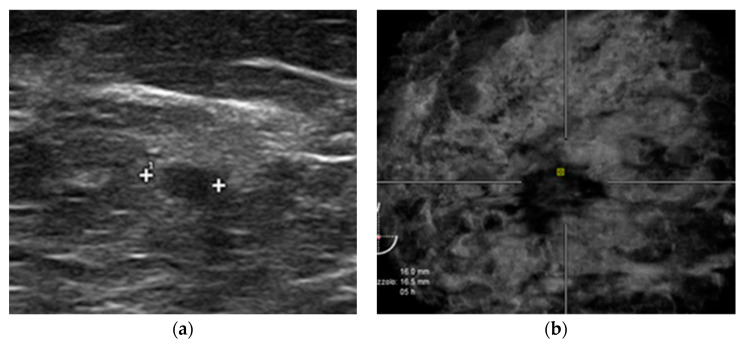
44-year-old woman comes to our institute for her first mammography. HHUS (**a**) shows a small nodule (triple negative invasive ductal carcinoma). Instead, there was a negative ABVS examination (**b**).

**Table 1 diagnostics-12-02170-t001:** Diagnostic performance (AUC, sensitivity, specificity, LR+) of HHUS and ABVS.

	HHUS	ABVS
Sensitivity	90.62%	68.75%
Specificity	96.32%	90.62%
LR+	24.6	7.33
AUC of ROC	0.930	0.788

**Table 2 diagnostics-12-02170-t002:** Main reasons for false-negative ABVS examination.

FN ABVS	Number of Exam
Peripheral nodules	5
Small aggressive cancer	2
Retro areolar findings	4
Clinical findings	2

## Data Availability

The datasets generated during and/or analyzed during the current study are available from the corresponding author on reasonable request.

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
