# Peer review of "A Prospective Comparative Evaluation of Handheld Ultrasound Examination (HHUS) or Automated Ultrasound Examination (ABVS) in Women with Dense Breast"

_diagnostics, 2022, doi:10.3390/diagnostics12092170_

Round 1

Reviewer 1 Report

- The name of 3D ulrasound is Automated Breast Ultrasound (instead of Automatic)

- In section 2.2. HHUS examination, row 75-78. The authors state that "The interpretation of mammography and the HHUS examination were performed 75 blindly from the results of ABVS." And after that, they mention "In all the patients, after HHUS, women underwent to ABVS". Should be corrected (probably the first statement).

- In section 2.2. HHUS examination, row 80, "both in sagittal and axial planes". HHUS is performed, according to the ACR BI-RADS lexicon in radial and antiradial planes, not axial and saggital.

- figure 5b needs adjustments.

- In the Discussion section, row 213-214, the authors state "In our knowledge, this is the first study in literature where ABVS had a worse diagnostic performance than HHUS". In their study, they mention that screening and symptomatic patients were included but they do not specify if the clinical data were available to the radiologists performing HHUS or interpreting ABUS. In our personal experience, HHUS has the great advantage that allows clinical examination and patient questioning before the performance of ultrasound. Maybe this could be an explanation for the author's results. However, the access to patients data should be mentioned.

Author Response

Dear Editor and Reviewers,

we would like to thank the Editor and the Reviewers for the thorough revision and help in improving the present manuscript. Please find below the changes that we made to improve the overall quality of the manuscript. We hope to have further improved both the readability of the manuscript and the scientific quality. We appreciated the constructive comments and we revised the manuscript according to the suggestions.

Reviewer#1:

Comment 1: “The name of 3D ultrasound is Automated Breast Ultrasound (instead of Automatic)

R: We thank the reviewer for the suggestion. Thus, we corrected as suggested.

Comment 2: In section 2.2. HHUS examination, row 75-78. The authors state that "The interpretation of mammography and the HHUS examination were performed 75 blindly from the results of ABVS." And after that, they mention "In all the patients, after HHUS, women underwent to ABVS". Should be corrected (probably the first statement).”

R: According to reviewers’ suggestion we removed the first statement. We would like to highlight that ABVS was done after HHUS but in different rooms and with different personnel. Than ABVS results (interpreted by pyisicians) was reported after ABVS image acquisition (made by radiology technicians and not physiscians). The physicians reporting HHUS were completely blind to ABVS results and viceversa.

Comment 3: “In section 2.2. HHUS examinationrow 80, "both in sagittal and axial planes". HHUS is performed, according to the ACR BI-RADS lexicon in radial and antiradial planes, not axial and saggital.” 

R: we thank the reviewer and we corrected as suggested. 

Comment 4: figure 5b needs adjustments”

R: We appreciated the comment of the reviewer and we try to improve the quality of the figure. This image shows a typical artifact of ABVS.

Comment 5: In the Discussion section, row 213-214, the authors state "In our knowledge, this is the first study in literature where ABVS had a worse diagnostic performance than HHUS". In their study, they mention that screening and symptomatic patients were included but they do not specify if the clinical data were available to the radiologists performing HHUS or interpreting ABUS. In our personal experience, HHUS has the great advantage that allows clinical examination and patient questioning before the performance of ultrasound. Maybe this could be an explanation for the author's results. However, the access to patients data should be mentioned.

R: R: we are grateful for the comment. the clinical data were available to the radiologists also in our study. According to the suggestion we mentioned this in the discussion row 216-219.

Reviewer 2 Report

In this study, the authors intend to compare the accuracy of the two methods, HHUS and ABVS. This is a very interesting topic. However, the different technicians who made the HHUS and ABVS diagnoses make it difficult to compare diagnostic accuracy. It is necessary to design a study that quantitatively evaluates the diagnostic accuracy of the two methods by the same technicians.

Although this is not the purpose of this paper, a qualitative comparison to the diagnostic results obtained by the two methods would be acceptable.

Author Response

Dear Editor and Reviewers,

we would like to thank the Editor and the Reviewers for the thorough revision and help in improving the present manuscript. Please find below the changes that we made to improve the overall quality of the manuscript. We hope to have further improved both the readability of the manuscript and the scientific quality. We appreciated the constructive comments and we revised the manuscript according to the suggestions.

Reviewer #2:

Comment: In this study, the authors intend to compare the accuracy of the two methods, HHUS and ABVS. This is a very interesting topic. However, the different technicians who made the HHUS and ABVS diagnoses make it difficult to compare diagnostic accuracy. It is necessary to design a study that quantitatively evaluates the diagnostic accuracy of the two methods by the same technicians.

Although this is not the purpose of this paper, a qualitative comparison to the diagnostic results obtained by the two methods would be acceptable.

R: we thank the reviewer for the suggestion. As reported in the manuscript, in the materials and methods section (rows 74-77), HHUS was performed by one medical radiologist whereas ABVS was performed by one technician dedicated to breast imaging, as it usually happens in Europe.

So, this data could be not exported to other realities (for example in the USA) where both exams were and are executed by technicians or sonologists. In our opinion, this is an important added value of HHUS especially in a clinical setting. Indeed, the radiologist can also perform patient questioning and clinical examination.  In addition, as you suggested, we highlighted further the differences between the two methods in the discussion section.

In our experience, we can say that both HHUS and ABVS are good tools with some potential in the screening setting because they can lead to an increase of the cancer detection especially in patients with negative mammography and dense breasts. In the United States reality, as reported by Wendie A. Berg, in the editorial “Current Status of Supplemental Screening in Dense Breasts” an important barrier to implementing screening ultrasound is manpower. For this reason, ABVS could be an important tool in the additional screening of breast cancer because the execution is carried out by the technician and the reading by radiologists. We added these considerations in the Discussion section.

Round 2

Reviewer 2 Report

After reviewing all the replies and the revised draft, I have changed my opinion.